# Analysis of Radiation Toxicity in Mammalian Cells Stably Transduced with Mitochondrial *Stat3*

**DOI:** 10.3390/ijms24098232

**Published:** 2023-05-04

**Authors:** Alisa Zanin, Giacomo Meneghetti, Luca Menilli, Annachiara Tesoriere, Francesco Argenton, Maddalena Mognato

**Affiliations:** Department of Biology, University of Padova, Via U. Bassi 58/B, 35131 Padova, Italy; alisa.zanin@studenti.unipd.it (A.Z.); giacomo.meneghetti@unipd.it (G.M.); luca.menilli@unipd.it (L.M.); annachiara.tesoriere@phd.unipd.it (A.T.); francesco.argenton@unipd.it (F.A.)

**Keywords:** UVC radiation, mitochondrial STAT3, genotoxic stress, cell survival, cell death

## Abstract

A coordinated action between nuclear and mitochondrial activities is essential for a proper cellular response to genotoxic stress. Several nuclear transcription factors, including STAT3, translocate to mitochondria to exert mitochondrial function regulation; however, the role of mitochondrial STAT3 (mitoSTAT3) under stressed conditions is still poorly understood. In this study, we examined whether the stable expression of mitoSTAT3 wild-type or mutated at the conserved serine residue (Ser727), which is involved in the mitochondrial function of STAT3, can affect the DNA damage response to UVC radiation. To address this issue, we generated mammalian cells (NIH-3T3 and HCT-116 cells) stably transduced to express the mitochondrial-targeted *Stat3* gene in its wild-type or Ser727 mutated forms. Our results show that cell proliferation is enhanced in mito*Stat3*-transduced cells under both non-stressed and stressed conditions. Once irradiated with UVC, cells expressing wild-type mitoSTAT3 showed the highest cell survival, which was associated with a significant decrease in cell death. Low levels of oxidative stress were detected in UVC-irradiated NIH-3T3 cells expressing mitoSTAT3 wild-type or serine-related dominant active form (Ser727D), confirming a role of mitochondrial STAT3 in minimizing oxidant cellular stress that provides an advantage for cell survival.

## 1. Introduction

The relationship between the nucleus and mitochondria is a challenging issue from different points of view, including gene expression regulation. Indeed, it is still not completely clear if nuclear transcription factors (TFs) found within mitochondria can regulate mitochondrial gene expression and/or affect mitochondrial function. STAT3 (signal transducer and activator of transcription 3) is a nuclear TF whose activity is involved in many different cellular processes, such as cell proliferation, survival, apoptosis, angiogenesis, immune response, and tumorigenesis [1,2,3]. The canonical role of STAT3 as a nuclear transcription factor requires phosphorylation at Tyr705, a conserved residue in its transactivation domain. STAT3 dimerization is activated by cytokines (i.e., IL-6) and growth factors (i.e., LIF, EGF) that, once bound by their receptors present on the plasma membrane, trigger STAT3 activation. STAT3 is activated by growth factor receptors with intrinsic tyrosine kinase activity through cytokine receptors associated with the Janus kinase (JAK) family tyrosine members and non-receptor tyrosine kinases (i.e., SRC) [4]. Once activated, dimers enter the nucleus and act as transcription factors by binding to DNA sequences. STAT3 can also act as a nuclear TF in a non-phosphorylated state by associating with nuclear factor-kB (NF-kB) and activating the transcription of genes involved in immune and oncogenic response [4,5].

In addition to the canonical nuclear role of STAT3, a mitochondrial role of STAT3 has been reported that requires phosphorylation at a conserved serine residue, Ser727 [6,7,8]. STAT3 localization inside mitochondria has been demonstrated [9], and its amount in mammalian mitochondria is 5–10% compared with that in the nucleus and cytosol and mainly regulates the activity of the electron transport chain [6,10]. In the last years, new data have emerged on the mitochondrial role of STAT3, showing its direct binding to mitochondrial DNA in mouse keratinocytes [11] and in mouse embryonic stem cells [12]. Recently, STAT3 has been demonstrated to interact directly with the mitochondrial scaffold protein prohibitin 1 (PHB1) and the tumor suppressor SH2D4 that is involved in blocking STAT3 nuclear translocation [13]. Recent evidence demonstrated that mitochondrially-localized STAT3 (mitoSTAT3) regulates the proliferation rate in zebrafish embryonic and larval stem cell niches through mitochondrial transcription [14]. While the involvement of nuclear STAT3 in response to cellular stress has been demonstrated [15,16,17], the role of mitoSTAT3 under stressed conditions is less known.

In this study, we analyzed the influence of mitochondrial STAT3 expression, in its wild-type and S727 mutated versions, on cellular response to DNA damage in mammalian cells. To this purpose, we created lentiviral constructs containing an engineered form of *Stat3* carrying wild-type or mutated sequence at S727, where serine replacement by alanine (S727A) impedes phosphorylation, functioning as a dominant negative, and serine replacement by aspartic acid (S727D) mimics a constitutively serine-phosphorylated STAT3 [9]. Lentiviruses were then used to infect mouse NIH-3T3 and human HCT-116 cells to generate stable cell lines expressing wild-type and mutated mitoSTAT3 protein, on which we investigated UVC radiation toxicity.

## 2. Results

### 2.1. MitoSTAT3 Can Modulate the Proliferation of NIH-3T3 Mouse Cells

We have generated mutant NIH-3T3 mouse cell lines stably expressing exogenous *Stat3* gene with wild-type (*MLS_Stat3_NES*) or mutated Ser-727 residue, with the substitution of the serine to an alanine (*S727A*, to block Serine phosphorylation) or aspartatic acid (*S727D*, to mimic the Serine phosphorylation) and with a mitochondrial localization sequence (MLS) and a nuclear export sequence (NES) that makes STAT3 unable to localize into the nucleus. As shown, the transduced cell lines showed the increased expression of *Stat3* mRNA (Figure 1a) and the increased amount of STAT3 protein (Figure 1b), compared to non-transduced cells. To further validate our lines, we have performed the immuno-fluorescent analysis of STAT3 protein subcellular localization, confirming that transduced cells accumulate STAT3 in mitochondria (Figure 1c).

Based on literature information, Serine 727 is fundamental for the mitochondrial STAT3-mediated proliferation of cells. To validate the role of this residue in our models, we have evaluated the number of cells at different time points after seeding (i.e., 24 h, 48 h, 72 h, 96 h). The growth curve of Figure 2a shows that cell numbers increased exponentially up to 72 h after seeding, and the cells transduced with wild-type or *S727D* mitochondrial *Stat3* gene more rapidly increased their number when compared to non-transduced cells or to cells transduced with mito*Stat3 S727A*. The doubling time of each cell line, calculated between 0 and 72 h after seeding, reported 3.7, 3, and 3.7 population doublings in MLS_Stat3_NES, MLS_Stat3_NES S727A, and MLS_Stat3_NES S727D cells, respectively, compared with 2.6 in non-transduced cells. The proliferation rate of each cell line was then evaluated by MTS assay in cells cultured for 48 h and 72 h. The results confirmed a significantly higher proliferation rate of mito*Stat3*-transduced cells than non-transduced cells at 48 h after seeding, which persisted significantly higher at 72 h after seeding for MLS_Stat3_NES cells (Figure 2c; *p* < 0.01). Cell cycle analyses carried out in cells recovered at these two time points after seeding showed a higher fraction of S-phase in MLS_Stat3_NES cells compared with other cells at 48 h after seeding (41%, Figure 2d). At 72 h after seeding, MLS_Stat3_NES cells showed a higher G_2_/M fraction than other cells (9% vs. 6%), and MLS_Stat3_NES S727D cells showed a higher fraction of S-phase cells than other cells (30% vs. 25%). To further analyze the proliferation ability of the mitoSTAT3-expressing cells, we performed the clonogenic assay. The results clearly indicate that the cloning efficiency of cells stably expressing mitochondrial STAT3 was higher than that of non-transduced cells, with a statistical difference for MLS_Stat3_NES and MLS_Stat3_NES S727D cells (Figure 2e; *p* < 0.01).

### 2.2. S727 Is Involved in the mitoSTAT3-Mediated Increase of Cell Viability upon UVC Stress

We examined the impact of the stable expression of mito*Stat3*, wild-type or mutated at Ser727, on NIH-3T3 cell response to UVC radiation by analyzing cell survival and apoptosis induction. The results of the MTS assay performed 24 h after irradiation with UVC 5 J/m^2^ indicate a higher viability of cells stably expressing wild-type or S727D mitoSTAT3 than non-transduced cells (*p* < 0.05, Figure 3a). We further evaluated cell viability by means of clonogenic assay, which measures the cycling ability of viable and healthy cells. Figure 3a shows that once irradiated, cells stably expressing mitoSTAT3 formed more colonies than control non-transduced cells, and the clonogenic survival of mito*Stat3*-transduced cells was significantly higher than that of non-transduced cells (57% in MLS_Stat3_NES, 36% in MLS_Stat3_NES S727A, and 51% in MLS_Stat3_NES S727D cells vs. 21% in control cells, *p* < 0.01; *p* < 0.05). To evaluate whether these differences were associated with changes in cell cycle progression, we performed cytofluorimetric analyses at 4 and 24 h after irradiation with UVC 5 J/m^2^. No significant differences were detected at 4 h after irradiation in the fraction of G_0_/G_1_-S-G_2_/M cells (Figure 3c). At 24 h after irradiation, the G_0_/G_1_ fraction decreased in MLS_Stat3_NES cells compared to an increase in the fraction of S- and G_2_/M-phase cells; nevertheless, these changes were not statistically significant (Figure 3d).

We analyzed whether the high clonogenic survival of irradiated mito*Stat3*-transduced cells was associated with a low level of cell death by determining the apoptotic index and activation of caspases 3 and 9. The fraction of apoptotic cells at 24 h after irradiation with UVC 10 J/m^2^ was 2.5% in non-transduced cells, 2.3% in MLS_Stat3_NES, 2% in MLS_Stat3_NES S727A, and 1.6% in MLS_Stat3_NES S727D cells, values significantly higher with respect to the non-irradiated counterparts (*p* < 0.01, *p* < 0.05, Figure 3e). Notably, in UVC-irradiated MLS_Stat3_NES S727D cells, the percentage of apoptotic cells was significantly lower than in non-transduced cells (*p* < 0.05). Mito*Stat3*-transduced cells showed a lower fraction of apoptotic cells with respect to non-transduced cells also without UVC irradiation, with a significant difference for MLS_Stat3_NES S727A and MLS_Stat3_NES S727D cells (*p* < 0.05, *p* < 0.01). The activation of caspase-3 was detected by fluorometric analyses in UVC-irradiated cells, showing a ~3-fold increase in non-transduced and MLS_Stat3_NES cells and a ~2-fold increase in MLS_Stat3_NES S727A and MLS_Stat3_NES S727D cells (Figure 3f). Caspase-9 was also activated in UVC-irradiated cells showing a 1.5-fold increase in non-transduced cells and a 0.7–0.9-fold increase in mito*Stat3*-transduced cells. The value of caspase-9 activation was significantly lower for MLS_Stat3_NES and MLS_Stat3_NES S727D cells compared with non-transduced cells (*p* < 0.05).

### 2.3. Oxidative Stress Induction and Mitochondrial Respiration in mitoStat3-Transduced Cells Irradiated with UVC

We evaluated the generation of reactive oxygen species (ROS) in living cells irradiated with UVC (10 J/m^2^) as a measure of oxidative stress. The results of the DCFH-DA assay showed that upon irradiation, ROS generation significantly increased in all cells (*p* < 0.01, *p* < 0.001, Figure 4a). In MLS_Stat3_NES and MLS_Stat3_NES S727D cells, the values of DCFH fluorescence were significantly lower compared with control non-transduced cells (*p* < 0.01, *p* < 0.05), whereas MLS_Stat3_NES S727A cells showed an ROS level similar to that of non-transduced cells. We then assayed cell respiration of mito*Stat3*-transduced cells by measuring oxygen consumption rate (OCR) changes in living cells, non-irradiated and irradiated with UVC. Figure 4b shows the results of the Seahorse mitochondrial stress test assay in non-irradiated cells, in which the different cell populations were sequentially injected with the ATPase inhibitor oligomycin (which decreases the OCR), an uncoupler agent (FCCP) (which increases oxygen consumption), and two electron transport chain inhibitors (rotenone/antimycin A) that shut down mitochondrial activity. As shown in the Seahorse bioenergetics map, under non-stressed conditions, MLS_Stat3_NES S727D cells displayed the highest rate of basal respiration, which was significantly higher than that of other cells (i.e., ~100 pmol/min vs. ~50 pmol/min). Furthermore, ATP-linked respiration was significantly higher in MLS_Stat3_NES S727D cells with respect to other cells (*p* < 0.0001), indicating for these cells a high activity of oxidative phosphorylation. Upon UVC irradiation, the bioenergetic response of MLS_Stat3_NES S727D cells was slightly modified, but still showed a higher ATP-linked respiration respect to non-transduced cells (*p* < 0.05). In irradiated MLS_Stat3_NES cells, the rate of basal respiration increased, but with variable values, making the increase insignificant. In non-transduced and MLS_Stat3_NES S727A cells, we detected an increase, not significant, of basal respiration and ATP-linked respiration compared with non-irradiated conditions.

### 2.4. Gene Expression Analysis in mitoStat3-Transduced Cells Irradiated with UVC

We measured by qRT-PCR the expression level of five genes having a central role in the DNA-damage response pathway: *Tp53*, *Cdkn1a* (*p21*), *Gadd45a*, *Ddb2*, and *Fdxr*. As shown in Figure 5 the *Tp53* gene showed slight alterations at both 4 and 24 h after UVC irradiation without evident differences among samples. *p21* was the most responsive gene against UV stress, showing a ~4.5-fold in mitoStat3-transduced cells and ~3-fold in non-transduced cells at 4 h after irradiation, which increased to ~6-fold at 24 h after irradiation, except in MLS_Stat3_NES S727D cells where it remained ~3-fold. However, the variable trend of *p21* expression did not produce statistically relevant values. Interestingly, at 4 h after irradiation, *Gadd45a* showed a significantly lower transcriptional response in mito*Stat3*-transduced cells than non-transduced cells (2.3-fold vs. ~1.3-fold, *p* < 0.05 vs. MLS_Stat3_NES; ** *p* < 0.01 vs. MLS_Stat3_NES S727A; *** *p* < 0.001 vs. MLS_Stat3_NES S727D). At this time point after irradiation, *Ddb2* showed a slight, but significant, induction in MLS_Stat3_NES S727A cells compared with other cells (1.25-fold vs. ~0.5-fold, * *p* < 0.05; ** *p* < 0.01). *Fdxr* was also induced at 4 h after IR (3-fold in non-transduced cells, 1.8-fold in MLS_Stat3_NES, 2.5-fold in MLS_Stat3_NES S727A, 2-fold in MLS_Stat3_NES S727D cells) with a significant difference between MLS_Stat3_NES and MLS_Stat3_NES S727D cells over non-transduced cells (** *p* < 0.01; *** *p* < 0.001). At 24 h after irradiation, all the analyzed genes, except *p21*, showed no relevant alterations in their expression level.

### 2.5. UVC Radiation Toxicity in mitoStat3-Transduced Human HCT-116 Cells

Human HCT-116 cells were stably transduced with mito*Stat3* (Appendix A) and analyzed for cell viability after UVC irradiation. The results of the MTS assay showed a significantly higher viability of mito*Stat3*-transduced than non-transduced cells cultured in 2D conditions, either without UVC or with UVC (Figure 6a). The results of the live–dead assay, which measures simultaneously live and dead cells based on intracellular esterase activity and plasma membrane integrity, showed a significantly lower fraction of dead cells in mitoStat3-transduced than non-transduced cells (12–14% vs. 19%, respectively, Figure 6b). The clonogenic assay showed a higher cloning efficiency of mito*Stat3*-transduced cells than non-transduced cells in both non-irradiated and irradiated conditions, with HCT-116 MLS_Stat3_NES cells showing a significant difference (*p* < 0.05, Figure 6c).

We also evaluated UVC radiation toxicity in mito*Stat3*-transduced HCT-116 cells cultured in 3D by using the LIVE/DEAD viability/cytotoxicity assay (Appendix A). Non-irradiated spheroids showed a bright green signal on their surface, indicating the viable cells, as the cells in this region have more nutrient and oxygen accessibility in contrast to the inner core, less oxygenated and less fed, which showed a red fluorescence. Once irradiated with UVC, spheroids still showed the bright green signal on their surface and the red signal in their inner core. Few dead cells were visible at the periphery of UVC-irradiated spheroids derived from non-transduced cells, in contrast to the evident presence of dead cells at the periphery of spheroids treated with methanol. UVC-irradiated spheroids derived from mito*Stat3*-transduced cells also showed a slightly detectable red fluorescence at their periphery.

## 3. Discussion

STAT3 activation by stress triggers a cellular response that is coordinated by genes involved in proliferation and survival pathways, together with cytokines and chemokines [17,18,19]. STAT3 serine 727 phosphorylation is fundamental for the maximal activation of STAT3-dependent transcription [20], but it is also associated with the mitochondrial function of STAT3 [8]. The ability of mitoSTAT3 to regulate the transcription of mitochondrial DNA [11,14] was confirmed here in NIH-3T3 cells stably transduced with wild-type mito*Stat3*, where we observed an increase in mitochondrial-encoded genes *Mt_nd1* and *Mt_nd4l* (Appendix A). The mitochondrial function of STAT3 in response to cellular stress has been studied in the area of ischemia/reperfusion injury and in response to UV radiation [18,21,22]. As reported in the literature, DNA damage induced by UVC, the most dangerous radiation among UV, consists mainly of DNA adducts (i.e., cyclobutane pyrimidine dimers) and, to a lesser extent, 6–4 photoproducts [23]. UVC-induced DNA damage activates the DNA damage response (DDR) pathway to repair DNA lesions by the nucleotide excision repair (NER) system or to induce apoptosis via intrinsic pathways when the damage is unrepairable [24]. In addition, UVC light activates growth factor receptors triggering the MAP-kinase cascade, which has a great impact on cell survival or apoptosis [25].

In the present study, we successfully generated mammalian cells stably transduced with mitochondrial *Stat3* and analyzed the contribution of mitoSTAT3 in the cell response against UVC radiation. We observed a significantly high proliferation rate of NIH-3T3 cells stably transduced with wild-type mitoSTAT3 (MLS_Stat3_NES Figure 2c,e), which was associated with a low endogenous level of apoptosis (Figure 3e). Interestingly, a similar increase in proliferation rate—associated with a low level of apoptosis—was also detected in cells with serine-related dominant active form (MLS_Stat3_NES S727D), confirming the important role of Serine 727 in the mito*Stat3*-dependent cell proliferation. Following UVC, cell viability and cloning efficiency of cells stably expressing MLS_Stat3_NES and MLS_Stat3_NES S727D were significantly higher than non-transduced cells (Figure 3a,b), showing the relevance of Ser727 in response to UV stress. On the whole, cytofluorimetric analysis reported slight perturbations in the cell cycle progression of irradiated cells, and this depends on the dose used (5 J/m^2^), which is lower than that of 20 J/m^2^, reported to activate a G_1_-phase arrest in these cells [26]. In response to UVC 10 J/m^2^, mito*Stat3*-transduced and non-transduced cells showed a significant increase in apoptotic cells (Figure 3e). However, the fraction of apoptotic cells was generally lower for both mito*Stat3*-transduced and non-transduced cells (≤2.5%), consistent with the fibroblastic nature of these cells that, in response to UV radiation, undergo replicative senescence rather than apoptosis in a dose-dependent manner [27,28]. UVC elicits apoptosis via an intrinsic pathway, which triggers mitochondrial outer membrane permeabilization, cytochrome c release, and subsequent caspase-9 activation through the apoptosome complex [29]. Caspase-9, once activated, cleaves the downstream caspases, including caspase-3. Our results showed that both caspase-3 and -9 activation occurred in UVC-irradiated cells, and that cells transduced with the wild-type form or the mutated *Stat3 S727D* showed a significantly lower activation of caspase-9 compared with non-transduced cells (Figure 3f). The role of S727 in response to UV stress was confirmed here by the reduced level of apoptosis and the reduced caspase-9 activation in MLS_Stat3_NES S727D cells compared with non-transduced cells.

The reduced apoptosis evidenced by our analysis, as correlated with S727, is also associated with reduced ROS production after UVC stress. ROS measurements with fluorescent probe 2′,7′-dichlorodihydrofluorescein can suffer as known from multiple artifacts and can represent a limit to the study. However, we detected a significantly lower ROS generation in cells transduced with mito*Stat3* wild-type or with *Ser727D* than in non-transduced cells and in cells with *Stat3 S727A* mutation (Figure 4a). The lower ROS production confirms the role of mitoSTAT3 in minimizing oxidant cellular stress [21], possibly due to the partial blocking activity of electron flow through complex I exerted by mitoSTAT3, which can protect mitochondria during stress [10]. The mitochondria of living cells play a key role in ROS formation, either under physiological or stressed conditions. Analysis of mitochondrial respiration showed that under non-stressed conditions, MLS_Stat3_NES S727D cells exhibited the highest basal respiration and ATP-linked respiration than other cells (Figure 4b,c), indicating their capacity to generate ATP via oxidative phosphorylation. In our live in vitro experiments, bioenergetic changes induced by UVC radiation were slightly affected compared with non-irradiated conditions. However, irradiated MLS_Stat3_NES cells increased their levels of basal respiration, whereas MLS_Stat3_NES S727D cells retained high levels of ATP-linked respiration. Since higher mitochondrial respiration is associated with higher cellular energy production, we suggest that MLS_Stat3_NES and MLS_Stat3_NES S727D cells respond better to genotoxic insult.

Upon irradiation with UVC, we found that STAT3 protein was induced in non-transduced cells but not in mito*Stat3*-transduced cells, most likely due to its endogenous overexpression in such cells (Appendix A). We, therefore, examined if this evidence was related to alterations in the expression of genes having a key role in the DDR pathway. To this aim, we analyzed, *Tp53*, *Cdkn1a (p21)*, *Ddb2*, and *Gadd45a*, which are targets of nuclear STAT3, and, in particular, *Tp53* and *p21* are direct STAT3 target genes [30], together with *Fdxr*, a well-known radiation-responsive gene, which codifies for a mitochondrial flavoprotein important for ROS-induced apoptosis in response to DNA damage [31,32]. P53 protein is required for safe cell response to genotoxic agents, among which UV radiation, by modulating the transcription of many genes, including *Cdkn1a* (p21), *Gadd45a*, and *Ddb2* [33,34,35]. The expression of p53 is inhibited at both transcriptional and protein levels in a Stat3-dependent manner thanks to a STAT3 binding site in the promoter of the *Tp53* gene [36]. At the same time, p53 directly affects *Stat3* mRNA [37], thus contributing to the complex response to DNA damage. In mito*Stat3*-transduced and non-transduced NIH-3T3 cells irradiated with UVC, we did not observe relevant changes in p53 mRNA level (Figure 5), in agreement with other studies [38,39,40]. Overall, our results of gene expression indicate that in response to UVC 5 J/m^2^, a transcriptional response occurs, mainly at the *p21* and *Fdxr* genes. *Cdkn1a* (*p21)* was induced at 4 h after irradiation in mito*Stat3*-transduced cells (~4.5-fold) and 2.8-fold in non-transduced cells), then, at 24 h after irradiation, *p21* resulted ~3-fold induced in MLS_Stat3_NES S727D cells and ~6-fold induced in MLS_Stat3_NES cells, MLS_Stat3_NES S727A, and non-transduced cells. P21 is the main protein triggering cell-cycle arrest in response to DNA damage by blocking G_1_-S and G_2_/M transitions through the inhibition of cdk2/cdk4 [41] and cdk1 [42], respectively. The induction of *p21* mRNA here detected after UVC was not associated with a cell cycle arrest in G_1_-S-phase. The functional role of p21 in suppressing cell proliferation during the cellular response to stress is not universal since p21 induction can also correlate with enhanced cell survival [43]; moreover, the cell cycle arrest after UV can be p21 independent [44]. Interestingly, the expression level of *Fdxr* was induced at 4 h after UVC, in agreement with previous findings [45], and to a significantly lower extent in MLS_Stat3_NES and MLS_Stat3_NES S727D cells than in MLS_Stat3_NES S727A and non-transduced cells. After UVC stress, *Gadd45a* was significantly down-regulated in mito*Stat3*-transduced cells compared to non-transduced cells, whereas *Ddb2* was significantly induced only in *MLS_Stat3_NES S727A* cells. Nevertheless, the observed little transcriptional responsiveness of *Gadd45a* and *Ddb2* genes, implicated in the repair of UVC-induced DNA damage by the NER pathway [46,47], is probably attributable to the low dose of UVC used in the present study (5 J/m^2^), which has a significant effect on cell viability and cloning efficiency but little effect on the expression of such genes that are responsive to high UVC doses (30–50 J/m^2^) [48,49].

Analysis of UVC radiation toxicity in human HCT-116 cells confirmed a significantly higher proliferation rate of mito*Stat3*-transduced than non-transduced cells under non-stressed conditions (Figure 6a). Once irradiated with UVC, mitoS*tat3*-transduced cells showed a significantly higher cell viability and a significantly lower death compared with non-transduced cells (Figure 6a,b). When HCT-116 cells were cultured in 3D, we observed less susceptibility at twice the dose of UVC used in 2D (i.e., 10 J/m^2^, Appendix A), possibly due to the lower number of dividing cells in the 3D spheroids (mostly localized at their periphery), in agreement with the results observed in other studies [50,51]. Nevertheless, dead cells were visible at the periphery of UVC-irradiated spheroids derived from non-transduced cells and slightly or not visible in those derived from mito*Stat3*-transduced cells (Appendix A). HCT-116 cells are *p53* wild-type, do not overexpress *Stat3*, and have been analyzed here to obtain information about radiation toxicity in terms of cell survival and cell death. No further investigations on ROS production and mitochondrial respiration were carried out on mito*Stat3*-transduced HCT-116 cells due to their tumoral status, which is associated with higher levels of ROS than normal cells [52,53].

Taken together, our results show that NIH-3T3 and HCT-116 cells stably transduced with mito*Stat3* have enhanced cell growth either under non-stressed conditions or in response to genotoxic stress exerted by UVC. Once irradiated, both NIH-3T3 and HCT-116 cells expressing wild-type mitoSTAT3 (MLS_Stat3_NES) showed the highest clonogenic survival, which was associated with a significant decrease in cell death. In response to UVC irradiation, NIH-3T3 cells expressing wild-type mitoSTAT3 showed the highest levels of basal mitochondrial respiration and a significant decrease of ROS production, confirming the role of mitoSTAT3 in minimizing oxidant cellular stress that advantages cells for survival. UVC-irradiated NIH-3T3 cells stably expressing S727D mutation showed lower ROS production and higher levels of basal respiration than non-transduced cells, which were associated with a significant increase in cell survival and decrease in cell death. Conversely, cells expressing S727A mutation exhibited ROS levels and respiration rates similar to non-transduced cells but showed a higher clonogenic survival than non-transduced cells. Here, based on the different behavior of the two mutant forms of STAT3, namely S727A and S727D, S727 phosphorylation is the relevant one for the response to UVC radiation. Nevertheless, account must be taken of the fact that STAT3 can undergo other post-translational modifications in addition to S727 residue [8] that ultimately could dictate its behavior in a more complex way. Notably, in this study, mito*Stat3*-transduced cells do possess nuclear *Stat3*; therefore, we considered the stress response of cells stably expressing mitoSTAT3 and nuclear STAT3 as well. For this reason, our results indicate that mitochondrial STAT3 can contribute to cellular resistance against genotoxic stress in the presence of nuclear STAT3.

## 4. Materials and Methods

### 4.1. Cell Cultures and Reagents

The human HEK-293T cells (epithelial from embryonic kidney, ATCC**^®^** CRL-3216^TM^) and the murine NIH-3T3 cells (from embryo, ATCC**^®^** n. CRL-1658^TM^) were cultured in DMEM high glucose, GlutaMAX™, pyruvate (ThermoFisher Scientific, Waltham, MA, USA) supplemented with 10% heat-inactivated fetal bovine serum (FBS, BIOCHROM, Berlin, Germany), and antibiotics streptomycin (50 µg/mL) and penicillin (50 U/mL) (ThermoFisher Scientific, Waltham, MA, USA). The human HCT-116 cells (colorectal cancer cells, ATCC**^®^** CCL-247 TM) were cultured in McCoy’s 5A Medium Modified (ThermoFisher Scientific, Waltham, MA, USA) supplemented with 10% FBS and antibiotics. Cells were incubated at 37 °C in a humidified atmosphere of 95% air and 5% CO_2_ and maintained in an exponential phase of growth by repeated trypsinization and reseeding prior to reaching sub-confluency.

### 4.2. Cell Irradiation

UVC irradiation was carried out in cell monolayers in Petri dishes (60 × 15 mm) by using a 254 nm UVS-11 mineral light lamp (Vilber Lourmat, Eberhardzell, Germany) at an average fluency rate of 0.25 mW/cm^2^ and exposure time was based on yield doses from 5 to 20 J/m^2^. The fluency of the UV light source was measured before each experiment with a UV radiometer (Vilber Lourmat, Eberhardzell, Germany). Cell medium was removed before irradiation and added back after irradiation. Cells were then processed at selected times after irradiation as indicated.

### 4.3. Generation of Lentiviral Constructs

Plasmids containing the murine *Stat3* cDNA flanked by a mitochondrial localization sequence (MLS) and a nuclear export sequence (NES) were constructed as previously reported [14]. Mutated forms of *MLS_Stat3_NES* mRNA were obtained from pCS2+MLS_Stat3_NES by site-directed mutagenesis using the Q5**^®^** Site-Directed Mutagenesis Kit (NEB). Construct insertion into lentiviral vectors has been carried out by recombination using GatewayTM LR clonaseTM II enzyme mix (Thermo Fisher). Lentiviruses were produced by transfecting lentiviral transfer and packaging plasmids into HEK-293T cells. Briefly, 24 h before, transfection cells (1.3 × 10^6^) were seeded in 10 × 100 mm culture dishes in complete medium w/o antibiotics. Third generation lentiviral vectors containing different forms of *Stat3*, packaging (Gag-Pol and Rev), and envelope (VSV-g) were incubated 20 min with polyethylenimine (PEI) according to the manufacturer’s protocol. Then, cells were incubated with the transfection reagent–plasmid mixture solution for 18 h. At the end of incubation time, culture medium was replaced with fresh complete medium without antibiotics, and 24 h later, the efficiency of transfection was tested by the lentiviral construct containing GFP. Then, 48 h after transfection, cell culture medium containing lentiviral particles was collected, centrifuged, and stored at −80 °C.

### 4.4. Generation of Stable Cell Line Expressing mitoStat3

Lentiviral particles were used to transduce NIH-3T3 and HCT-116 cells. Then, 24 h after seeding, the cells were incubated with lentivirus for 48 h; then, the transduction efficiency was determined by monitoring the fluorescence intensity of the green fluorescence from the control GFP under fluorescent microscope (Appendix A). The cells were then cultured in complete selective medium containing G-418 (450 µg/mL) for two weeks enabling us to select cells stably expressing wild-type mitoSTAT3 (*MLS_Stat3_NES*) and mutated mitoSTAT3 at serine 727 (*MLS_Stat3_NES S727A*, *MLS_Stat3_NES S727D*).

### 4.5. Cell Proliferation Assay

Cells stably transduced with wild-type or mutated mito*Stat3* and non-transduced cells were maintained in exponential phase during sub-cultures. For the growth curve, cells were seeded in different 60 × 15 mm culture dishes (5 × 10^3^ cells/cm^2^) in complete fresh medium at day 0 and then counted each of the following 4 days by trypan blue dye exclusion.

### 4.6. MTS Assay

For the MTS (3-(4,5-dimethylthiazol-2-yl)-5-(3-carboxymethoxyphenyl)-2-(4-sulfophenyl)-2H-tetrazolium salt) assay, cells (5 × 10^3^) were seeded in triplicate in 96-well plates in complete culture medium as previously reported [54,55]. After 24 h, cells were irradiated, incubated for 24 h, and then assayed for cell viability by the CellTiter 96 Aqueous One Solution Cell Proliferation Assay (Promega, Madison, WI, USA). In brief, the culture medium was removed, and the cells were incubated for 60–90 min in the dark with 20 µL of the MTS reagent diluted in 100 µL of serum-free medium. The adsorbance of the formazan product was recorded at 490 nm with a microplate reader (Spectramax 190, Molecular Device, San Jose, CA, USA). Cell viability was calculated by comparing the adsorbance values of irradiated vs. non-irradiated cells that were considered 100%.

### 4.7. Clonogenic Assay

The colony-forming assay cells were seeded in 60 × 15 mm culture dishes and allowed to attach for 24 h. Then, cells were harvested by trypsinization and counted by Trypan blue dye exclusion. Then, 200 viable cells were plated in 60 × 15 mm culture dishes in complete fresh medium and grown for 10–12 days. Cells were then stained with crystal violet (Sigma-Aldrich, St. Louis, MO, USA), and the cloning efficiency (CE) was calculated as the proportion of cells that originated colonies (formed by at least 50 cells) to the total number of cells plated, expressed in percentage. The CE values were then used to calculate clonogenic cell survival, expressed as the percentage of the CE of irradiated cells over that of non-irradiated cells (100%).

### 4.8. Live-Dead Assay

We used the LIVE/DEAD Viability/Cytotoxicity kit (Molecular Probes^TM^, Invitrogen, Eurogene, OR, USA) that allows us to distinguish between live and dead cells using two dyes, calcein AM (calcein acetoxymethyl ester) and ethidium homodimer-1 (EthD-1). The intracellular esterase activity of the live cells converted the virtually nonfluorescent cell-permeant calcein AM to the intensely fluorescent calcein. The polyanionic dye calcein is well retained within live cells, producing an intense, uniform green fluorescence in live cells (ex/em ~495 nm/~515 nm). EthD-1 is highly positively charged; it cannot cross cell membranes to stain living cells, it only enters cells with damaged membranes and undergoes a 40-fold enhancement of fluorescence upon binding to nucleic acids, thereby producing a bright red fluorescence in dead cells (ex/em ~495 nm/~635 nm). HCT-116 cells, stably transduced with mito*Stat3* and non-transduced, were seeded in 96-well plates and 24 h later were irradiated with UVC (10 J/m^2^). At 24 h after irradiation, cells were incubated for 30–45 min in the dark with 150 μL of a diluted solution containing the two dyes, EthD-1 and calcein AM, according to manufacturer’s indications. The fluorescent emission of EthD-1 and calcein AM was measured in a 96-well plate reader (EnVision, Perkin Helmer, Waltham, MA, USA), and the percentage of live and dead cells was calculated by comparing the fluorescence values of UVC-irradiated vs. non-irradiated cells that were considered as 100%.

### 4.9. Apoptosis Assay

Cells stably transduced with wild-type or mutated mito*Stat3* and non-transduced cells were seeded in 60 × 15 mm culture dishes (20 × 10^3^ cells/cm^2^) in complete fresh medium, incubated for 24 h and then irradiated with UVC. The activation of caspases 3 and 9 was measured by the Caspase-9 Fluorimetric assay kit (BioVision, Abcam, Waltham, MA, USA) at 24 h after irradiation. The fluorescent emission at λ = 505 nm (excitation at λ = 400 nm) of cleaved amino-4-trifluoromethyl coumarin (AFC) was measured in a 96-well plate reader (EnVision, Perkin Helmer). Apoptotic cells were recognized based on the fragmented morphology of their nuclei stained with 4,6-diamino-2-phenylindol (DAPI, Roche, Basel, Switzerland, 2 mg/mL), following standard protocol [54]. At least 500 cells/sample/experiment were scored under a LeicaDM4000 fluorescent microscope (Leica Microsystems, Wetzlar, Germany).

### 4.10. Flow Cytometry Analyses

Cell cycle analysis was carried out as previously described [56]. Briefly, 1 × 10^6^ cells stably transduced with wild-type or mutated mitoSTAT3 and non-transduced cells were recovered from non-irradiated and UV-irradiated cells, fixed in ethanol, and then stained with propidium iodide (PI, Sigma-Aldrich, 50 µg/µL) and DNase-free RNase (100 µg/mL, Thermo Scientific^TM^). After 1 h incubation at 37 °C, the samples were analyzed using a BD LSRFortessa X-20 flow cytometer (BD Biosciences, Franklin Lakes, NJ, USA); data from 25 × 10^3^ cells/sample were collected for acquisition and cell cycle distribution analysis using BD FACSDivaTM software version 9.0.

### 4.11. RNA Extraction and qRT-PCR

Cells stably transduced with wild-type or mutated mito*Stat3* and non-transduced cells were plated in 60 × 15 mm culture dishes in complete fresh medium and, 24 h after, irradiated with UVC. Then, total RNA was isolated by using Trizol**^®^** Reagent (Invitrogen, ThermoFisher Scientific, Waltham, MA, USA) according to the manufacturer’s protocol. Total RNA quantification was performed using the ND-1000 spectrophotometer (NanoDrop Technologies, Wilmington, DE, USA). For mRNA detection, 1 μg of total RNA was retrotranscribed with ImProm-II Reverse Transcription System (Promega), and qRT-PCR reactions were performed with the GoTaq qPCR Master Mix (Promega), as previously reported [57,58]. Reactions of qRT-PCR were carried out in triplicates according to the following amplification protocol: 95 °C for 2 min, 95 °C for 15 s, 60° for 1 min (40 cycles). Gene-specific primers for *Tp53*, *Cdkn1a (p21)*, *Gadd45a*, *Ddb2*, and *Fdxr* and for *Gapdh* as reference are available in Appendix A. The relative expression level of genes was calculated using the comparative delta Ct (threshold cycle number) method (2^−ddCt^) [59] implemented in the 7500 Real-Time PCR System software v1.5.1 (Applied Biosystems^TM^ 7500 Fast Real-Time PCR System, ThermoFisher Scientific, Waltham, MA, USA).

### 4.12. Cell Extracts and Western Blot

Cells stably transduced with wild-type or mutated mitoSTAT3 and non-transduced cells were plated in 60 × 15 mm culture dishes in complete fresh medium, then cellular extracts were prepared from UV-irradiated and non-irradiated cells at 4 and 24 h after irradiation. Whole cell extracts from 3 × 10^6^ cells were prepared in 200 µL RIPA lysis buffer, then we determined protein concentration and loaded equal amounts of protein for electrophoresis on 10% polyacrylamide precast gels (Invitrogen, Life Technologies, Carlsbad, CA, USA).

### 4.13. Immunofluorescence and Fluorescence Microscopy

Cells stably transduced with mito*Stat3*, wild-type or mutated, and non-transduced cells were seeded in 60 × 15 mm culture dishes containing glass coverslips (18 × 18 mm) for 24 h. Cells were then fixed and permeabilized, as previously reported [60]. Cells were incubated for 1 h at room temperature in blocking solution (PBS + Triton X-100 0.1% + goat serum 10%) and then incubated overnight at 4 °C with primary antibodies anti-Stat3 (Cell Signaling Technology, Danvers, MA, USA, 124H6) and anti-ATAD3 (Proteintech, Rosemont, IL, USA, 16610-1-AP) diluted 1:100 in PBS + 0.2% Triton X-100, 10% goat serum. At the end of incubation, cells were washed in PBS and once in PBS + 0.1% Triton X-100 and subsequently incubated at room temperature for 1 h with Alexa Fluor 488 goat anti-mouse secondary antibodies and Alexa Fluor 594 donkey anti-rabbit antibodies (Life Technologies, 1:250). Then, cells were washed, counterstained with 2 µg/mL DAPI (4,6-diamidino-2-fenilindole) in antifade solution (Vectashield, Vector Laboratories, Newark, CA, USA), and cover glass slips were mounted. The immunofluorescence images were captured with an epifluorescence microscope (Leica DM6 B, Wetzlar, Germany) equipped with the Leica DFC7000 T Camera and LAS X Software Leica Application Suite X 3.7.5.24914. Images were processed by Adobe PhotoShop 8.0 software (Adobe, San José, CA, USA).

### 4.14. Measurement of ROS Generation in Living Cells

Cells stably transduced with wild-type or mutated mito*Stat3* and non-transduced cells were seeded in quadruplicate in 60 × 15 mm dishes (20 × 10^3^ cells/cm^2^) in complete culture medium. Twenty-four hours after seeding, half dishes were irradiated with UVC (10–20 J/m^2^), then irradiated and non-irradiated dishes were processed for ROS generation by using the Dichlorodihydrofluorescein diacetate (DCFH-DA) probe (Invitrogen, Life Technologies).

### 4.15. Seahorse Assay

Cells (50 × 10^3^ cells/cm^2^) stably transduced with wild-type or mutated mito*Stat3* and non-transduced cells were seeded in Seahorse Cell Culture XF96 microplates (Agilent Technologies, Santa Clara, CA, USA) and incubated at 37 °C in a humidified cell incubator in 5% CO_2_ for 24 h. One of the two microplates seeded for each experiment was irradiated with UVC (10 J/m^2^), and oxygen consumption rate (OCR) was measured with a Seahorse assay in irradiated and non-irradiated microplates.

### 4.16. Establishing 3D Cultures

The 3D cell cultures of mito*Stat3*-transduced and non-transduced HCT-116 cells were established by seeding 2000 cells in 200 μL/well of complete culture medium in ultra-low attachment PrimeSurface® 96U plates (S-bio^TM^), and after 24 h we observed the spheroid organization. The plates are pre-coated with unique ultra-hydrophilic polymer that enables spontaneous 3D cell formation in a scaffold-free modality.

### 4.17. Statistical Analyses

The results are reported as means ± standard deviation (S.D.) or standard error (S.E.). The differences between groups were evaluated by two-tailed, unpaired Student’s *t*-test and differences with a *p*-value < 0.05 were considered significant.

## Figures and Tables

**Figure 1 ijms-24-08232-f001:**
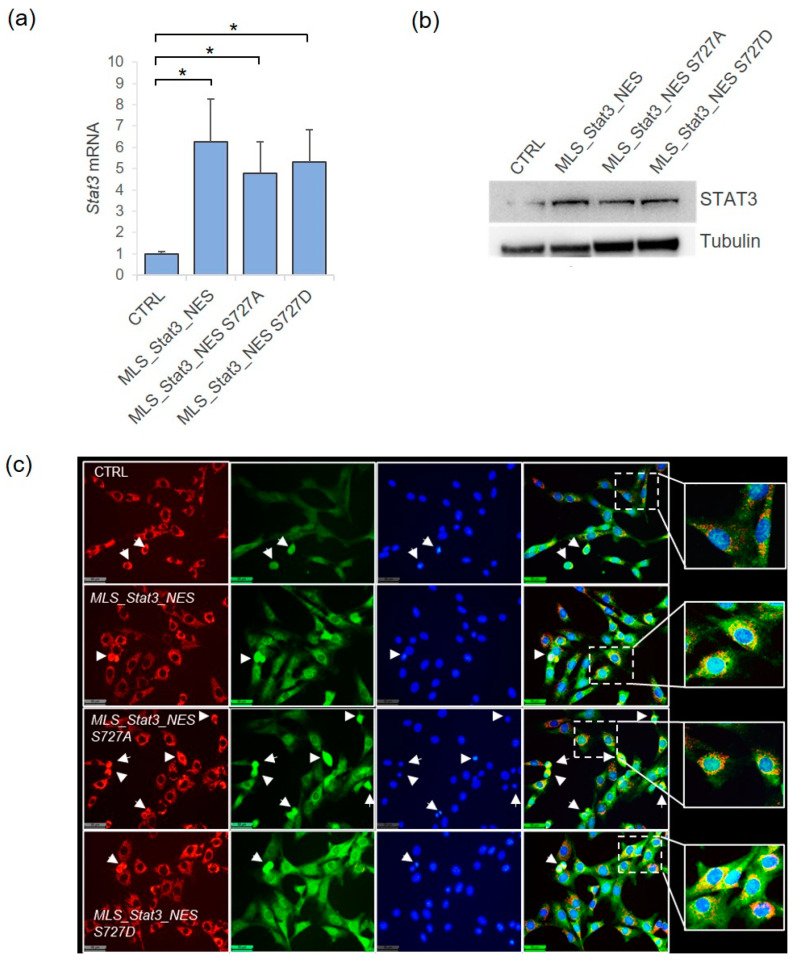
Analysis of STAT3 expression in NIH-3T3 cells stably transduced with mito*Stat3*. (**a**) The relative expression of *Stat3* mRNA was measured by qRT-PCR in non-transduced cells (CTRL) and in cells transduced with wild-type mito*Stat3 (MLS_Stat3_NES)* or with mutated Ser727; data are mean ± S.D. from three independent experiments (* *p* < 0.05, Student’s *t*-test). (**b**) STAT3 protein detected by Western blotting on whole cell lysates; tubulin was used as loading control. (**c**) Immunofluorescence analysis of STAT3 cellular localization in cells co-immunostained with anti-STAT3 (green) and anti-ATAD3 (red) as a mitochondrial marker. Cell nuclei are counterstained with DAPI (blue). The slight green fluorescence in nuclei is due to endogenous nuclear STAT3. The yellow signal of the merged red and green fluorescence is visible in mitochondria of interphase cells and in cells under mitosis (white arrows). Enlarged views of representative merged images are shown. Images were captured under a fluorescent microscope Leica DM6B (40×, scalebar represents 50 µm).

**Figure 2 ijms-24-08232-f002:**
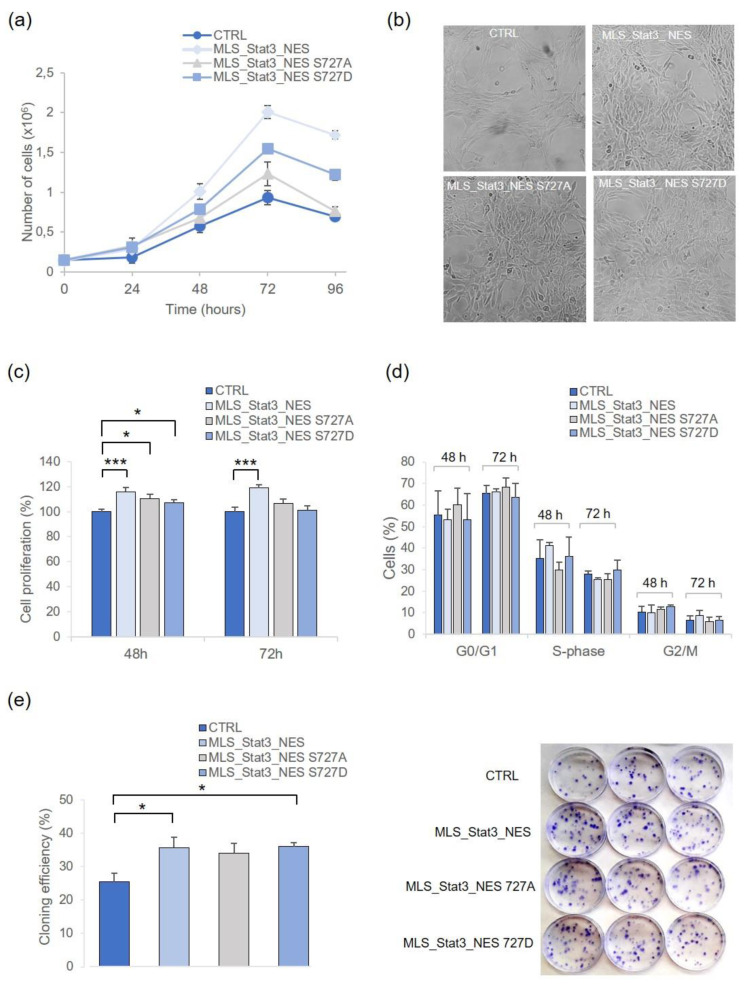
Cell proliferation is enhanced in mito*Stat3*-transduced NIH-3T3 cells. (**a**) Cell growth curve determined by Trypan blue dye counting at day 1, 2, 3, and 4 after seeding. Data refer to means ± S.D. of independent experiments (3 ≤ *n* ≤ 6). (**b**) Bright-field microscopy at 48 h after seeding; mito*Stat3*-transduced cells were maintained under selective medium containing G-418 (450 µg/mL). (**c**) Cell proliferation determined at 48 h and 72 h after seeding by MTS. Data are means ± S.E. from independent experiments (3 ≤ *n* ≤ 5; * *p* < 0.05, *** *p* < 0.001, Student’s *t*-test). (**d**) Cell cycle analysis by flow cytometry at the same time points; data were collected from 25.000 cells/sample using a BD LSRFortessa X-20 flow cytometer and mean ± S.D. from three independent experiments is indicated. (**e**) Clonogenic assay; data refer to means ± S.E. of independent experiments (3 ≤ *n* ≤ 5), each carried out in quadruplicate (* *p* < 0.05, Student’s *t*-test). Representative images of clones derived from mito*Stat3*-transduced and non-transduced cells are shown.

**Figure 3 ijms-24-08232-f003:**
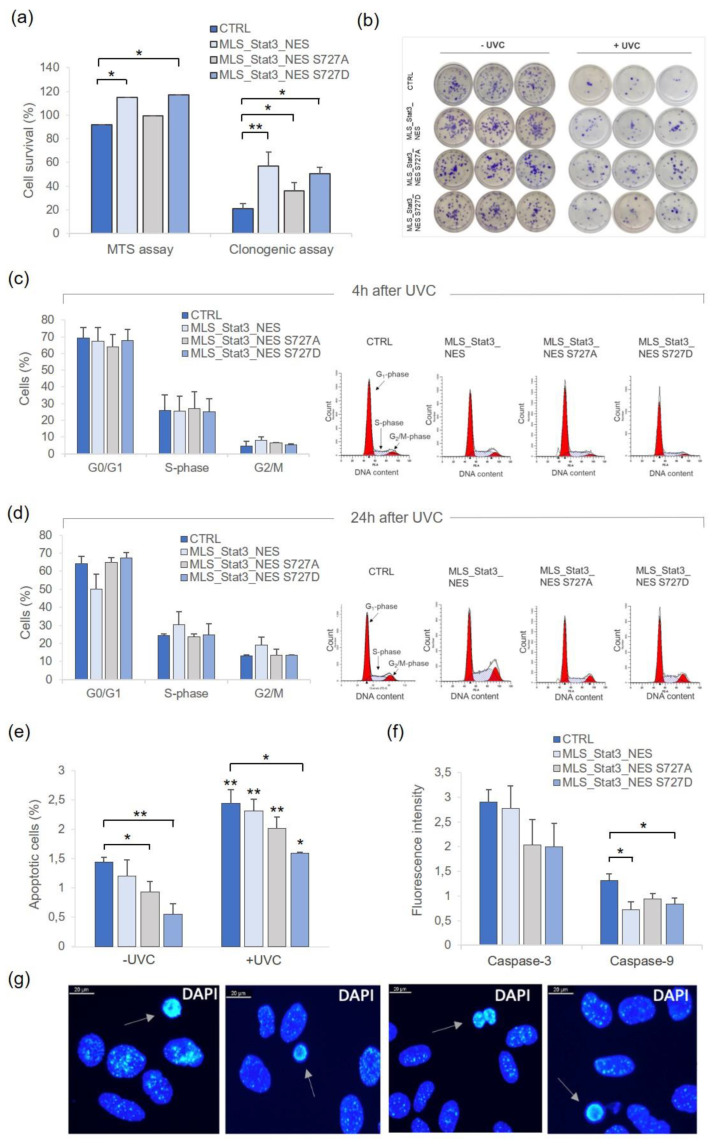
Cell survival and apoptosis induction in mito*Stat3*-transduced NIH-3T3 cells irradiated with UVC. (**a**) Cell survival determined by MTS assay and clonogenic assay at 24 h after UVC irradiation (5 J/m^2^). For MTS assay, data are mean ± S.D. from three independent experiments, each one carried out in triplicate (* *p* < 0.05, Student’s *t*-test). For clonogenic assay, data refer to means ± S.E. from 3 ≤ *n* ≤ 6 independent experiments, each carried out in triplicate (* *p* < 0.05, ** *p* < 0.01, Student’s *t*-test). Clonogenic survival of irradiated cells determined as the percentage of cloning efficiency of irradiated cells with respect to that of non-irradiated cells (expressed as 100%). (**b**) Representative images of clones derived from non-irradiated cells and irradiated cells (5 J/m^2^). (**c**,**d**) Cell cycle analyses were carried out by flow cytometry in irradiated cells recovered at 4 and 24 h after irradiation with UVC (5 J/m^2^). Data were collected from 25.000 cells/sample using a BD LSRFortessa X-20 flow cytometer. Data refer to means ± S.D. of three independent experiments. Representative plots of cell cycle analysis carried out at 4 and 24 h after irradiation are shown. (**e**) Apoptotic index determined by morphological analysis of DAPI-stained cells at 24 h after irradiation with UVC (10 J/m^2^); data refer to mean ± S.D. from three independent experiments (* *p* < 0.05; ** *p* < 0.01; Student’s *t*-test). (**f**) Activation of caspase-3 and -9 measured by fluorimetric analyses at 24 h after irradiation with UVC (10 J/m^2^). The data of fluorescence intensity are presented as fold-increase of irradiated over relative non-irradiated cells (mean ± S.D. from three independent experiments, * *p* < 0.05, Student’s *t*-test). (**g**) Representative images of UVC-irradiated cells stained with DAPI are shown. Apoptotic cells are indicated with arrows.

**Figure 4 ijms-24-08232-f004:**
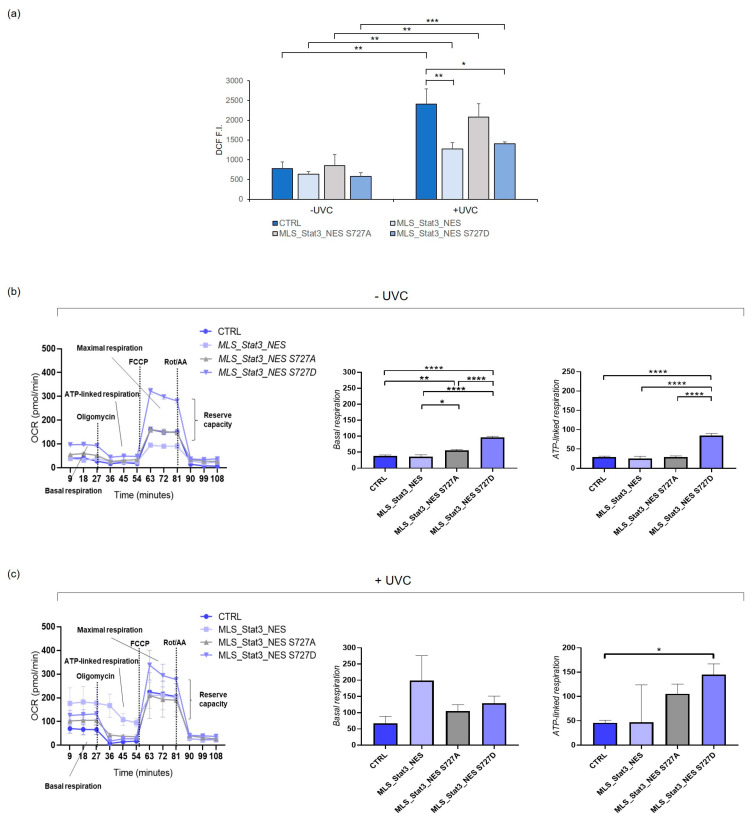
ROS generation and mitochondrial respiration in mito*Stat3*-transduced NIH-3T3 cells. (**a**) ROS production determined in non-irradiated and UVC-irradiated (10 J/m^2^) cells by the fluorescent probe 2′,7′-dichlorodihydrofluorescein (DCFH), administered in its diacetate form (DCFH-DA), which enters cells and then is oxidized to the fluorescent product 2′,7′-dichlorofluorescein (DCF). The fluorescence intensity (F.I.) of DCF has been determined by cytofluorimetric analysis. Data are presented as means ± S.D. from three independent experiments (* *p* < 0.05, ** *p* < 0.01, *** *p* < 0.001, Student’s *t*-test). Oxygen consumption rate (OCR) measurements determined by Seahorse XFp analyzer in non-irradiated (**b**) and UVC-irradiated (**c**) NIH-3T3 cells. Following plating, cells (50 × 10^3^ cells/cm^2^) were allowed to equilibrate for 1 h at 37 °C and then were injected with drugs at the time points indicated: oligomycin (2 mM), FCCP (p-trifluoromethoxy carbonyl cyanide phenyl hydrazone, 1 mM), and rotenone/antimycin A (0.5 mM). Basal respiration is the basal oxygen consumption rate measured in the cells prior to injection of oligomycin; ATP-linked respiration is the respiration that drives ATP synthesis, which is determined using sensitivity to the ATP-synthase inhibitor oligomycin; maximal respiration is the maximal rate of electron transport supported by mitochondria once the FCCP uncoupler has been injected. Bar charts of basal respiration and ATP-linked respiration of OCR plot are shown. The data are presented as means ± S.D. from three independent experiments (* *p* < 0.05, ** *p* < 0.01, *** *p* < 0.001, **** *p* < 0.0001, Student’s *t*-test).

**Figure 5 ijms-24-08232-f005:**
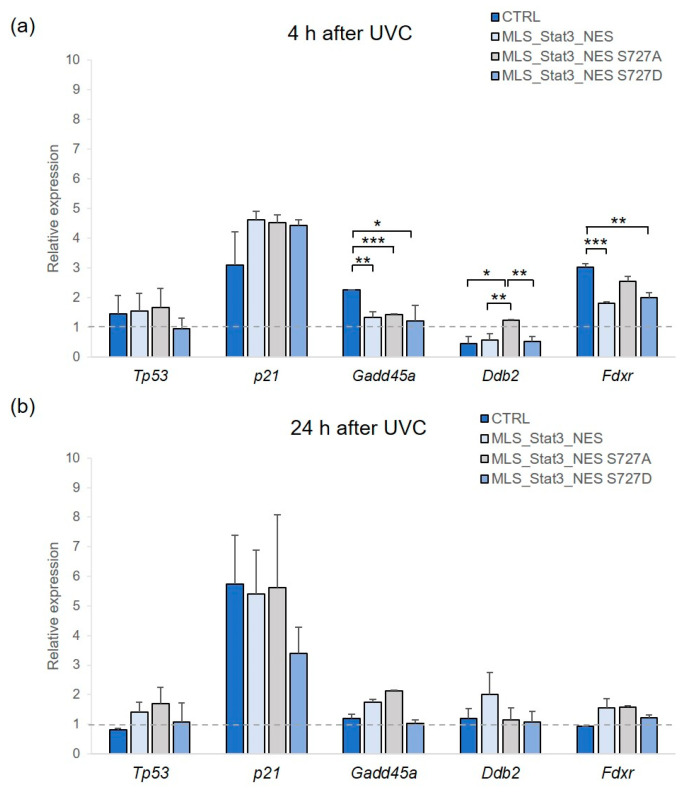
Gene expression analysis in mito*Stat3*-transduced NIH-3T3 cells irradiated with UVC. The relative expression of *Tp53*, *p21*, *Gadd45a*, *Ddb2*, and *Fdxr* mRNAs was analyzed by qRT-PCR at 4 (**a**) and 24 h (**b**) after UVC irradiation (5 J/m^2^) in mito*Stat3*-transduced and non-transduced cells. Bars represent the means ± S.D. of independent experiments (3 ≤ *n* ≤ 4), each performed in triplicate, expressed as fold-change of UVC-irradiated vs. non-irradiated cells (* *p* < 0.05; ** *p* < 0.01; *** *p* < 0.001, Student’s *t*-test). The dotted line represents the value “1” when no change is observed.

**Figure 6 ijms-24-08232-f006:**
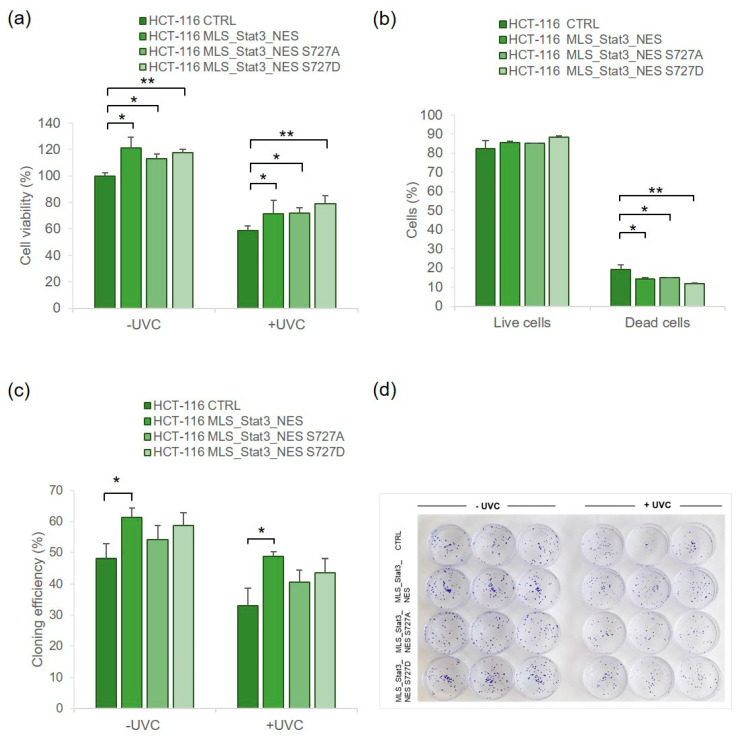
Analysis of UVC radiation toxicity in human HCT-116 cells transduced with mito*Stat3*. Cell viability determined by MTS assay at 24 h after UVC (5 J/m^2^) (**a**). Fraction of live and dead cells determined by live–dead assay at 24 h after UVC (5 J/m^2^) (**b**). Data are means ± S.D. from independent experiments, each performed in six technical replicates (* *p* ≤ 0.05, ** *p* ≤ 0.01, Student’s *t*-test). (**c**) Clonogenic assay performed in UVC-irradiated and non-irradiated cells. Data refer to means ± S.D. of three independent experiments, each carried out in triplicate. (* *p* ≤ 0.05, Student’s *t*-test). (**d**) Representative images of irradiated and non-irradiated HCTT-116 clones.

## Data Availability

The data presented in this study are available on request from the corresponding author M.M. upon reasonable request. The data are not publicly available due to privacy.

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
