# Peer review of "Analysis of Radiation Toxicity in Mammalian Cells Stably Transduced with Mitochondrial Stat3"

_ijms, 2023, doi:10.3390/ijms24098232_

Round 1

Reviewer 1 Report

In this manuscript by Zanin et al, the authors transduced NIH-3T3 and HC/-116 mammalian cells with a MLS-Sstat3_NES encoding lentiviral construct to express mitochondrial targeted STAT3 –WT, -S727A (to block Serine phosphorylation) or –S727D (to mimic Serine phosphrylation). These cells were then irradiated with UVC to induce oxidative stress and checked for cell proliferation, cell survival, induction of apoptosis and response to ROS. Their results clearly indicate that while the expression of STAT3-WT and –S727D minimized the UVC-induced oxidative stress, STAT3-S727A did not suggesting that phosporylation of mitochondrial STAT3 is crucial for a proper defense to UVC radiation toxicity.

This is an interesting paper that sheds more light on the role of STAT-3 at mitochondria. The authors’ conclusions are supported by a large number of replicates used for each experimental approach. My only minor concern regards few conclusions on the discussion section. Here (lines 308-312) the authors should state that, based on the different behavior of the 2 mutant STAT-3 forms (namely –S727D and S727A), is S727 phosporylation relevant for the response to UVC and not simply Ser727.

Minor points.

The description of the construct indicated in lines 58-61 should be moved to the first part of the  results section (i.e. lines 71-74)

Line 90. “foundamental” should be “fundamental”

Legend of Figure 2. “d” should be “e”. Therefore, the legend of “d” is missing

In the Materials and Methods the authors should describe the procedure used to calculate the relative expression.

Reviewer 2 Report

The paper investigates the effects of mitochondrial STAT3 phosphorylation on cancer cell growth.

I liked the idea of the research and the experimental design. The results obtained are important to understand the future perspective for cancer treatment. However, there are some issues with the manuscript:

1. Row 67. You mention generation of stable cell lines. Did you indeed generate stable cell lines, or just use transduced cells maintained on the antibiotics?

2. Fig 1b. Is there statistical significance between control and other points? Moreover, why do you mark the cells with mitosis in Fig 1c?

3. Row 102. “The linear part of the growth curve was used to calculate the doubling time of each cell line…” There is no linear part in an exponential curve.

4. Fig.2. Multiple issues with the figure legend: no description for 2(e); also check the whole text once more.

5. Please, reformulate the subtitle 2.2. (rows 123-124).

6. The English needs to be improved throughout the text.

7. There is no description of qRT-PCR in the Methods.

8. ROS measurement with Dichlorodihydrofluorescein diacetate suffers from multiple artifacts. It should be noted as a limitation of the study.

Reviewer 3 Report

Nuclear transcription factor Start3 is present in the mitochondria and functions in mitochondrial respiration. This paper investigated the role of mitochondrial Stat3 on the DNA damage response to UVC radiation. The mechanism underlying protective role of exogenous Start3 expression against UVC irradiation is unclear. Authors need to address this reviewer’s concerns which are described below.

Major issues

1.     Authors concluded that exogenous expression of Stat3 protein is present in mitochondria in Figure 1. However, Stat3 is expressed in the nuclei in contrast to ATAD3. Authors should precisely explain the result.

2.     Introduction of Stat3 into the cells may enhance respiratory activities of mitochondria. Authors should measure mitochondrial complex activity or mitochondrial membrane potential.

3.     Statistical significance was not shown in cell cycle analysis (Figure 2d, 3c, 3d). Significant change in distribution of cell cycle was not observed among indicated cells.

4.     UVC irradiation induced apoptosis in all cells. Percentage of apoptotic cells is less than 2.5%. The mechanism underlying protective role of exogenous Start3 expression against UVC irradiation is unclear.

5.     Explain the mechanism of increase in survival rates of cells expressing mtSTAT3 S727A mutation. These cells showed no change of ROS levels and respiration rate compared to non-transduced cells.

6.     In figure 6b, basal levels of percentage of dead cells without UVC irradiation were not shown.

7.     Supplemental figures are not provided.

Minor issue

Figure legend of figure 2d is incorrect.

Round 2

Reviewer 3 Report

The authrds have answered all my concerns